# Do Different Sutures with Triclosan Have Different Antimicrobial Activities? A Pharmacodynamic Approach

**DOI:** 10.3390/antibiotics11091195

**Published:** 2022-09-03

**Authors:** Frederic C. Daoud, Fatima M’Zali, Arnaud Zabala, Nicholas Moore, Anne-Marie Rogues

**Affiliations:** 1INSERM, BPH, U1219, Université de Bordeaux, 33000 Bordeaux, France; 2UMR 5234 CNRS, Laboratoire de Microbiologie Fondamentale et Pathogénicité, Université de Bordeaux, 33000 Bordeaux, France; 3Pôle de Santé Publique, Service d’Hygiène Hospitalière, CHU Bordeaux, 33000 Bordeaux, France

**Keywords:** suture, antimicrobial, pharmacodynamics, triclosan, surgical site infection, time-kill, contact killing, translational modelling

## Abstract

(1) Background: Three antimicrobial absorbable sutures have different triclosan (TS) loads, triclosan release kinetics and hydrolysis times. This in vitro study aims to analyse and compare their antimicrobial pharmacodynamics. (2) Methods: Time-kill assays were performed with eight triclosan-susceptible microorganisms common in surgical site infections (SSIs) and a segment of each TS. Microbial concentrations were measured at T0, T4, T8 and T24 h. Similar non-triclosan sutures (NTS) were used as controls. Microbial concentrations were plotted and analysed with panel analysis. They were predicted over time with a double-exponential model and four parameters fitted to each TS × microorganism combination. (3) Results: The microbial concentration was associated with the triclosan presence, timeslot and microorganism. It was not associated with the suture material. All combinations shared a common pattern with an early steep concentration reduction from baseline to 4–8 h, followed by a concentration up to a 24-h plateau in most cases with a mild concentration increase. (4) Conclusions: Microorganisms seem to be predominantly killed by contact or near-contact killing with the suture rather than the triclosan concentration in the culture medium. No significant in vitro antimicrobial pharmacodynamic difference between the three TS is identified. Triclosan can reduce the suture microbial colonisation and SSI risk.

## 1. Introduction

There is a broad array of surgical wound closure methods, including thousands of suture types, staples and surgical adhesives [1]. Sutureless surgery is also being developed in various fields, including maxillofacial and cardiac surgery [2,3,4,5]. Minimising the risk of surgical site infection (SSI) is an important consideration when developing surgical wound closure techniques.

Triclosan is a synthetic, hydrophobic bisphenol (5-Chloro-2-(2,4-dichlorophenoxy)phenol) [6]. It is solid below 54 °C and displays low solubility in water (10 µg/mL in pure water at 20 °C) compared to nonpolar solvents such as olive oil (approximately 600,000 µg/mL) or ethanol (>1 million µg/mL) [7]. Hydrolysis and photodegradation are the two main triclosan degradation pathways. Both are too slow to be measurable over 24 h, providing the assays are protected from intense light. Triclosan has several properties that make it a broad-spectrum antimicrobial, especially its nonpolarity, which brings triclosan molecules together, among other nonpolar substances such as phospholipids influences bacterial cell membranes [8,9,10,11]. Triclosan lipid-membranotropism facilitates its concentration inside cell phospholipids and the membranes of gram-positive cocci (mix of peptidoglycan and phospholipids) and gram-negative bacteria (predominantly phospholipids). It also partially explains triclosan’s lower ability to penetrate the outer walls of *C. albicans*, predominantly consisting of polysaccharides [12]. Triclosan’s antimicrobial activity has multiple targets, but the main one is reported to be NADH-dependent enoyl-[acyl carrier protein] reductase (FabI). This inhibits cell membrane fatty acid synthesis, thus disrupting membranes [13,14].

Cell membrane lipidic composition is not enough to explain triclosan susceptibility. For example, *Pseudomonas aeruginosa* cell membrane is predominantly lipidic but has an enzymatic membrane efflux pump sufficient to expel triclosan, thus reducing its concentrations and antimicrobial effects [15,16].

Triclosan is added to absorbable sutures to inhibit microbial colonisation and thus reduce the risk of SSI [17,18,19,20,21,22]. Braided polyglactin-910 sutures are available with a maximum triclosan load of 472 µg/m (V+) [23]. Monofilament polydioxanone (P+) and monofilament poliglecaprone 25 sutures with a maximum of 2360 µg/m (M+) [23,24,25]. These sutures are also available without triclosan (V, P and M).

A previous study analysed the triclosan release kinetics of V+, P+ and M+ in pure static water and accelerated conditions calibrated to reproduce subcutaneous and intramuscular release in operated large animals [26]. It established the relation between the triclosan release rate and the antistaphylococcal activity in V+. However, the pharmacodynamics of V+, P+ and M+ once implanted in live operated tissues are not documented.

Several surface static agar cultures have measured zones of inhibition (ZOI) of sutures with triclosan (TS) vs. non-triclosan controls (NTS). One assay showed that V+ inhibited the growth of *Staphylococcus aureus* and *Staphylococcus epidermidis* after 24-h exposure, while V caused no inhibition [27]. Others showed similar results with M+ vs. M and P+ vs. P. *Escherichia coli* was inhibited up to approximately 1 cm from P+ [28,29,30]. Those in vitro experiments also showed the inhibition of methicillin-resistant strains of *S. epidermidis*, *S. aureus* and *Klebsiella pneumoniae*. In vivo experiments in one study showed that subcutaneous TS segments in mice with 7 × 10^6^
*E. coli* colony-forming units (CFU) inoculum displayed, when removed after 48 h, a 90% microbial reduction, while NTS controls were colonised [28]. The same study showed in guinea pigs with a 4 × 10^5^
*S. aureus* CFU inoculum a 99.9% microbial reduction.

While these studies confirm triclosan’s antibacterial activity, they do not demonstrate the translation of the results to operated human tissues regarding triclosan bioavailability and TS antimicrobial activity. The level, duration and volume of antimicrobial activity around TS are uncertain.

The fact that 88% (22/25) (11,957 patients) of parallel-arm prospective randomised controlled clinical trials (RCT), which are the most comprehensive meta-analyses published to date, are non-significant supports translational uncertainty, although the pooled relative risk (RR) was 0.73 [0.65, 0.82] [31]. The WHO has published a conditional guideline recommending the use of TS to reduce the risk of SSI, stating that the quality of the evidence is moderate [32].

Many factors influence SSI risk, including the suture material, microbial concentration, infected volume, microbial multiplication, susceptibility to triclosan, exposure duration, surgical site characteristics and patient’s natural defences [33].

The objective of this study (AD16-174/AST2016-181/IIS15-216/2016-11-09) was to analyse the in vitro pharmacodynamics of the three TS, understand their translational antimicrobial characteristics and compare them (Appendix A).

## 2. Materials and Methods

### 2.1. Microbiology: Time-Kill Assays

Nine microorganisms common in SSIs were selected, representing a range of triclosan minimum inhibitory concentrations (MIC). Those were *E. coli* ATCC 25922 (MIC 0.03 µg/mL), *E. coli* ESBL producer collection clinical strain (MIC 0.03 µg/mL), *S. epidermidis* CIP 8155T (MIC 0.03 µg/mL), *S. aureus* ATCC 29213 (MIC 0.03 µg/mL), Methicillin-resistant *S. aureus* (MRSA) ATCC 33592 (MIC 0.03 µg/mL), MRSA collection clinical strain, *Candida albicans* ATCC 10231 (MIC 4 µg/mL), *C. albicans* collection clinical strain and *P. aeruginosa* collection clinical strain (MIC 256 µg/mL). V, V+, P, P+, M and M+ challenged all the microorganisms. All sutures had USP 2-0 calibres, i.e., a 0.35 to 0.399 mm diameter, a 35 cm length and, thus, a 0.03 to 0.04 cm^3^ volume.

A time-kill assay protocol was specified according to CLSI standards [34]. Sutures and microbial cultures were handled in a safety cabinet. Sutures were unpackaged and immediately cut into various lengths, including 35 cm-long segments. Suture incubation on agar plates for 18 h at 37 °C checked sterility.

Agar plate cultures each microorganism’s purity. A single colony was inoculated in a 10 mL sterile tryptic soy culture broth (TSB; Difco, BD Diagnostics, Sparks, MD, USA) tube. The suspension cultures were incubated for 4 h at 37 °C with continuous shaking until exponential growth was reached with 0.5 McFarland turbidity (approximately 1.5 × 10^8^ colony-forming units CFU/mL). A densitometric controlled inoculum of the culture was extracted. It was added to 10 mL of the culture medium in a sterile tube setting at an approximately 10^6^ CFU/mL baseline microbial concentration. After testing different segment lengths, 35 cm was selected because it enabled the distinguishing of concentration differences between each timeslot. Each culture had one immersed segment incubated for 24 h at 37 °C. Microbial concentrations were determined at four timeslots: baseline, 4, 8 and 24 h (T0, T4, T8, T24).

Culture tubes underwent 4 min of 48 Mhz sonication to detach viable microorganisms from the tube walls and suture. A 100 µL sample was drawn and then underwent five serial 1/10 *v*/*v* dilutions. The original sample and dilutions were spread on separate agar plates with Mueller Hinton (MH) medium. The plates were incubated for 24 h at 37 °C and used if colonies were countable. Plate colony count multiplied by the dilution factor was the source culture microbial concentration (CFU/mL). Six copies of the 54 sutures × microorganism combinations were performed.

### 2.2. Data Analysis

#### 2.2.1. Plots of Time-Kill Assays

For each combination and each timeslot (T0 through T24), the viable microbial concentration was converted to log_10_(CFU/mL). Repeated time-kill assays were plotted jointly.

#### 2.2.2. Statistical Analysis

A comprehensive panel analysis tested the association of microbial concentration with four factors (1). This model included TS, NTS and *P. aeruginosa*.
(1)$$C\left(t\right)=b_{0}+b_{1}x_{triclosan.i}+b_{2}x_{material.i}+b_{3}x_{microorganism.i}+b_{4}x_{timeslot.i}\;$$

A second-panel analysis focused only on TS and triclosan-sensitive microorganisms (2).
(2)$$C\left(t\right)=b_{0}+b_{1}x_{material.i}+b_{2}x_{microorganisms.i}+b_{3}x_{timeslot.i}\;$$

The *.i* indices refer to the dummy variables defined for each modality of each independent factor. The models were calculated using random effects (RE) panel regression if the Hausman test for random effects was non-significant and in the presence of the heteroskedasticity of microbial concentration residues, confirmed by a significant Breusch–Pagan Lagrange multiplier test (LM). If those conditions were not met, the model would be calculated using the pooled ordinary least squares (OLS) method [35].

Specific findings underwent exploratory post hoc tests.

#### 2.2.3. Pharmacodynamic Fitting of Microbial Concentration

Triclosan bactericidal and antifungal activities and microbial multiplication are two competing dynamics. Therefore, a predictive two-term model was used to predict observations (3).
(3)$$C\left(t\right)=C_{1}e^{-tL_{1}}+C_{2}e^{tL_{2}}\;$$

*C*(*t*) is the microbial concentration in the tube at time t from baseline. The first term describes the microbial decrease over 24 h, with *C*_1_ solved using *C*(0)/2, i.e., the half of the mean microbial concentration among the six repetitions at baseline. It is equal to the inoculum divided by the culture medium volume. *Ln*2/*L*_1_ is the half-life (*T*1_1/2_) of microbial killing over 24 h. The negative exponent defines an exponential decay.

The second term describes microbial growth, with *C*_2_ solved using *C*(0)/2. *Ln*2/*L*_2_ is the half-life (*T*1_1/2_) over 24 h. The implicit positive exponent defines the exponential growth.

The four unknown parameters were solved jointly given the observed datasets consisting of (time, microbial concentration) couples for each TS × strain combination and each repetition. The solution targeted minimising the least-squares of observations vs. the predicted values with the following constraints: (1)The targeted adjusted coefficient of determination *R²_adjusted_* (4) was within [0.9, 1] and(2)The predicted concentration at T24 was equal to the observed mean concentration at T24 among the six repetitions. *n* was the number of observations per combination (*n* = 4 measurements × 6 repetitions = 24 per combination), and *k* was the number of estimated explanatory parameters in the function using the *n* observations (*k* = 4, per combination).
(4)$$R_{adjusted}^{2}=1-\left(1-\frac{\sum_{i=1}^{n}{\left(y_{i}-{\hat{y}}_{i}\right)}^{2}}{\sum_{i=1}^{n}{\left(y_{i}-\bar{y}\right)}^{2}}\right)\frac{n-1}{n-\left(k+1\right)}$$

The *R²_adjusted_* measures the goodness-of-fit (GoF) of the model and thus the degree of prediction of observations. An *R²_adjusted_* equal to 1 indicates the perfect prediction of the observed concentrations, while an *R²_adjusted_* equal to 0 shows that the model does not predict any better than random guesses.

Two required initial inputs (the partial concentrations at T0: *C*_1_ and *C*_2_) were defined such that *C*_1_*e*^0^ + *C*_2_*e*^0^ = *C*(0), given *e*^0^ = 1.

Data management and statistical analyses were performed with the xt module in Stata 17, StataCorps LLC, College Station, TX, USA.

The fitting of the pharmacodynamic model to the data was performed in Microsoft Excel 2019 version 2205, Microsoft Corporation, Redmond, WA, USA, using the Solver function. Parameter resolutions were checked with Maple 2021.1, Maple Inc., Waterloo, ON, Canada.

## 3. Results

### 3.1. Time-Kill Assays

The plots show growth with the eight triclosan-sensitive strains with NTS control sutures (Figure 1a–c, Figure 2a–c, and Figure 3a,b) and *P. aeruginosa*, which was used as the triclosan-resistant control strain (Figure 3c).

The plots with TS and triclosan-sensitive strains show an initial rapid microbial concentration reduction between baseline and T4 or T8. The reduction magnitude ranges between about 1.5 log_10_ and 3 log_10_. It is followed by a microbial concentration plateau between T8 and T24. The plateau ranges between a mild decline, a steady concentration and a mild increase.

### 3.2. Statistical Analysis–Comprehensive Panel Model

The test eligibility criteria applied to the comprehensive model indicate the applicability of OLS regression (Table 1). The adjustment of the model is moderate, given R² = 0.578. The regression coefficients of the model are large and significant when comparing TS to the NTS controls, when comparing timeslots to baseline T0 and when comparing the *P. aeruginosa* triclosan-resistant control to triclosan-sensitive *S. epidermidis*, the reference level in the model. All other coefficients have a small magnitude and are non-significant. This shows that the microbial concentration is significantly associated with the presence or absence of triclosan, timeslots and *P. aeruginosa.* Suture materials and other microbial strains are not associated with microbial concentration in this comprehensive model.

### 3.3. Statistical Analysis–Focused Panel Model

The test eligibility criteria applied to the focused model indicated the use of random-effects panel regression (Table 2). The adjustment of the model is improved with R² = 0.687. The regression coefficients of the model are large and significant when comparing the timeslots to the T0 baseline and are significant but mild when comparing the 5 triclosan-sensitive microbial strains to *S. epidermidis* as a reference level for microorganisms in the model. The coefficients for suture materials have a small magnitude and are non-significant. This model shows that the microbial concentration is significantly associated with timeslots, mildly associated with some microbial strains and not associated with suture materials.

### 3.4. Statistical Analysis–Post Hoc Paired t-Test

The mean microbial reduction among triclosan-susceptible microorganisms, from baseline (T0) to trough (T4 or T8), was −2.29 log_10_ of CFU/mL [–2.40, –2.19].

The post hoc paired t-test showed a mild but significant mean concentration increase from trough to T24 (+0.36 log_10_ of CFU/mL [+0.26; +0.46], *p* < 0.0001).

### 3.5. Pharmacodynamic Fitting of Microbial Concentration

Each of the 24 combinations was fitted with a predictive pharmacodynamic function. Examples with *S. epidermidis* are shown in Figure 4, *S. aureus* in Figure 5, *E. coli* in Figure 6, and examples with ESBL-producing *E. coli* in Figure 7. Subfigures (a), (b) and (c) are fittings with V+, P+ and M+, respectively.

The predictive pharmacodynamic functions of the 24 combinations, based on a common algebraic function and fitted parameters, are listed in Table 3. The GoF, estimated by the adjusted *R²_adjusted_*, is between 0.61 and 0.89 in 22 fitted functions. There is a poor fit in the two other functions (*R²_adjusted_* between 0.38 and 0.49) with *C. albicans ATCC 33529* and the monofilament sutures P+ and M+ (Appendix A).

All functions show an early fast microbial concentration decline from baseline to T4 or T8 h, followed by a plateau until T24. The plateau has a steady concentration in 2 cases (8.33%), a mild concentration decrease in 4 cases (16.7%) and a mild concentration increase in 18 cases (75%).

## 4. Discussion

### 4.1. Protocol Specifications and Interpretation

This study performed the first in vitro pharmacodynamics analysis of TS antimicrobial activity using time-kill assays. V, P, M and triclosan-resistant *P. aeruginosa* were the controls. The experimental settings were the same as those used in static water release kinetics [26].

The suspension cultures had enough volume and nutrients to sustain microbial growth beyond the T24 timeslot, as confirmed by the growth in the NTS cultures (Figure 1 to Figure 3). Microbial colony count was proportional to the microbial concentration in the cultures, with a degree of random error between repetitions. 

Microbial concentrations exceeding the 10^8^ CFU/mL upper boundary could not be accurately estimated, but that had no impact on the analysis. None of them reached the lower detection boundary below 10^2^ CFU/mL.

### 4.2. Key Results

The plots and statistical analyses showed that the microbial concentration is significantly associated with triclosan, the timeslot and the microorganism. It is not associated with the suture material.

The plots with all TSs and triclosan-susceptible microorganisms consisted of an initially rapid microbial reduction from T0 to T4 or T8, with a mean reduction of −2.29 log_10_ of CFU/mL followed by a plateau without a microbial concentration change, mild decrease or mild increase. The mean change between the T8 and T24 was a mild but significant increase.

The underlying mechanisms of the predictive pharmacodynamic models were assumed to be microbial kill and multiplication. The fitting was good in all combinations except for two. The models differed little between TS types for a given microorganism. Most differences were between microorganisms. The functions reproduced the initial microbial concentration rapid reduction and subsequent plateau and showed a mild increase in 75% of combinations.

### 4.3. Interpretation of The Time-Kill Assays

These assays show the ability of V+, P+ and M+ to reduce a high microbial concentration over a 4-to-8-h period, with very little difference given each microorganism despite the 5-fold difference in the triclosan load and suture structure. The plateau and frequent mild increase are unexpected.

The key question is whether the predominant killing was by the triclosan concentration in the culture medium or by the contact killing at the suture surface or near it, where the triclosan concentration is high. Indeed, triclosan is a hydrophobic solid whose dissolution follows the Noyes & Whitney principle, which explains the triclosan gradient between the solid surface and the bulk of the solvent in a static tube. This gradient disappears when the tube is shaken. Therefore, its dissolution rate is slow around the suture, whose volume is 0.3 to 0.4% of the culture volume. Therefore, unless the tube is shaken, the triclosan diffusion layer forms a gradient with a maximum close to 10 µg/mL at the suture surface, decreasing with the distance from the source [7,26,36,37,38,39].

The culture conditions in static TSB (water + 3% organic nutrients and minerals) were closer to pure static water release kinetic determinations than to the ethanol/water 3.3% w/w solution with 24 rounds-per-minute constant rotation [26]. The triclosan release in 10 mL of pure static water is with V+ 2.3, 3.3 and 6.8 µg at 4, 8 and 24 h, respectively, with P+, 5.3, 6.7 and 6.1 µg and with M+, 7.6, 9.6 and 7.4 µg [26]. After sonication, the triclosan concentrations were homogeneous in the bulk of the cultures, i.e., with V+ 0.23, 0.33 and 0.68 µg/mL at 4, 8 and 24 h, respectively, with P+, 0.53, 0.67 and 0.61 µg/mL and with M+, 0.76, 0.96 and 0.74 µg/mL. 

The triclosan minimal bactericidal concentrations (MBC) of *S. aureus* (0.03 to 2 µg/mL), *E. coli* (0.03 to 16 µg/mL) and *C. albicans* (0.12 to 16 µg/mL) are presented [40,41]. Therefore, the triclosan concentrations were within the MBC ranges, as the first sonication. This should have caused microbial killing through T24 in all TS ×microbial combinations. The plateau and a mild increase in 75% of combinations after T8 suggest that the triclosan concentration in the medium was too low to prevent microbial multiplication.

The available data do not provide proof of the mechanism. However, a potential explanation is that the dispersed triclosan after the T4 and T8 sonications were captured in the lipids of the killed bacteria. Therefore, the plateau between T8 and T24 is likely to be an experimental artefact when the tubes are still, and the two terms of the pharmacodynamic functions offset each other.

The obtained in vitro data do not show why the mild microbial concentration increased during the T8–T24 plateau in 75% of the assays. One potential explanation is the gradual decrease in the release rate while exponential microbial growth continued.

### 4.4. Comparison with Other Preclinical Studies

These in vitro pharmacodynamic models and underlying release kinetics are compatible with the absence of *E. coli* and *S. aureus* surface growth on agar plates with TS segments explanted from rodents after 48 h [26,28]. 

One study attempted to measure the duration and level of TS antimicrobial activity with a TS segment transferred consecutively from one static surface agar culture plate to another for up to 30 days [42]. The conclusions were that TSs display antimicrobial activity from about one week to one month depending on the TS × microorganism.

The methods of these in vitro experiments must be considered when interpreting the results. (1) The two-dimensional diffusion around the TS segments on the agar surface cultures represents, at most, the amount that would be contained in the three-dimension diffusion layer of the suspension culture. (2) The water/air surface diffusion meets less resistance than it does in full immersion. Therefore, ZOI overestimates, by several-fold, the antimicrobial volume of TS. (3) The TS triclosan release in static cultures is 15 to 60 times slower than that observed in large animal subcutaneous or intramuscular explants [26]. Therefore, the in vitro antimicrobial activity duration is also overestimated.

### 4.5. Translational Interpretation to Live Operated Human Tissues

The translational application of this pharmacodynamic study to operated human tissues is limited.

(1) The TS release rate in the time-kill assays is 15 to 60 times slower than it is in operated tissues. (2) Surgical sites are much larger than 10 mL tubes, so the suture volume is much less than 0.3 to 0.4% of the surgical site. (3) The permanent in vivo motion maintains a thin diffusion layer around the sutures, so the contact or near-contact volume with drifting microorganisms is probably negligible compared to the surgical site volume at risk. (4) When natural defences are functional, scattered bacteria are rapidly killed, and few encounter the TS. When natural defences are weak, scattered microbial multiplication requires antibiotics and/or reintervention

Therefore, preventing microbial colonisation while the triclosan release rate is efficacious, i.e., a few hours after implantation, is the only result of this pharmacodynamic study that can translate to a surgical site. However, that can relieve natural defences by reducing the risk of microbial colonisation of the suture. The three TS types share similar in vitro antimicrobial activity. There is no indication they would have significantly different in vitro antimicrobial activities in operated tissues. Measurements of microbial dose or concentration reduction cannot translate to operated tissues because they do not consider the complexity of surgical sites and natural defences.

## 5. Conclusions

This in vitro study shows that triclosan sutures kill susceptible microorganisms that come in direct contact or near contact with their surface. The in vitro antimicrobial profiles of the braided polyglactin-910, monofilament polydioxanone and monofilament poliglecaprone 25 sutures present no significant difference, and no difference in the operated tissues is predicted.

The in vitro pharmacodynamics suggest a significant reduction in the microbial dose close to the sutures as of implantation. Triclosan can minimise the suture colonisation risk early on, relieve natural defences and reduce the risk of surgical site infection.

## Figures and Tables

**Figure 1 antibiotics-11-01195-f001:**
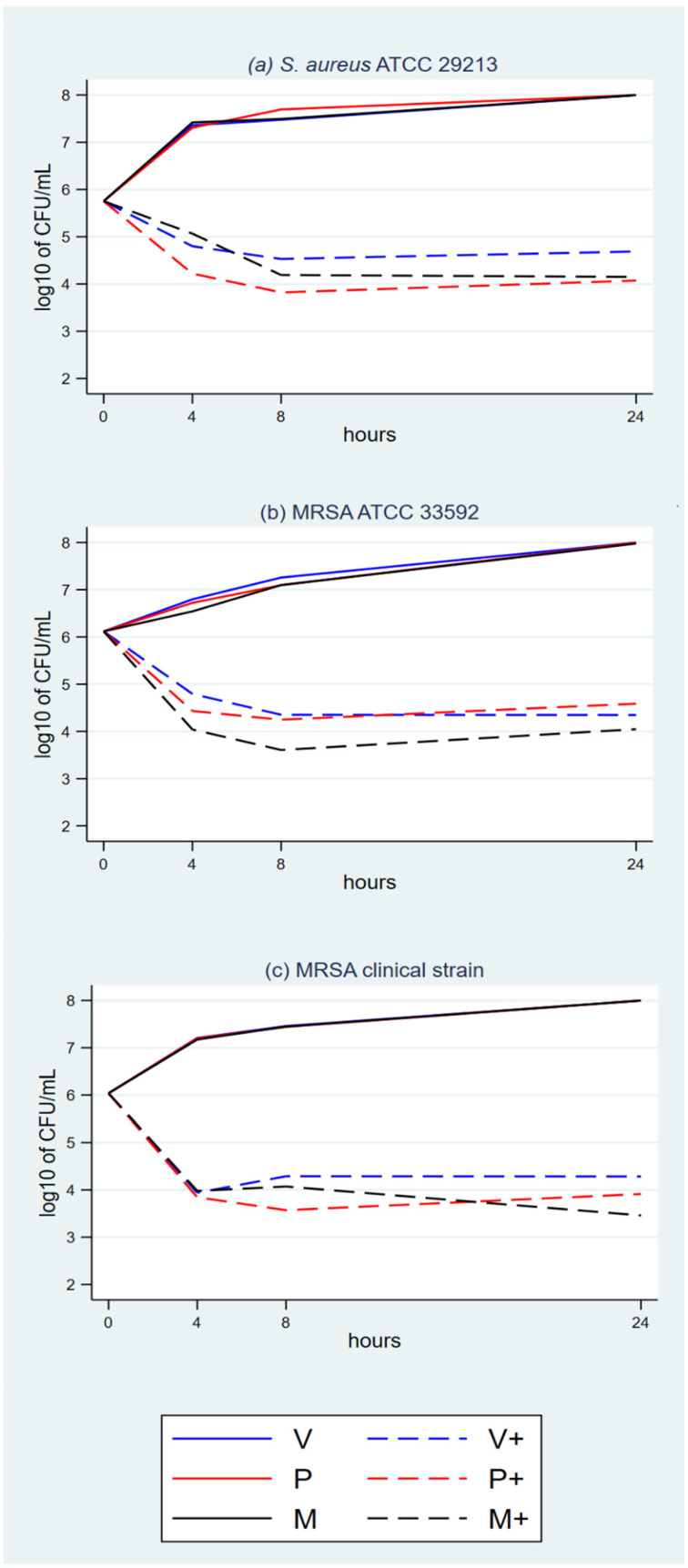
(**a**–**c**) Time-kill analyses *S. aureus* and MRSA with the three TS.

**Figure 2 antibiotics-11-01195-f002:**
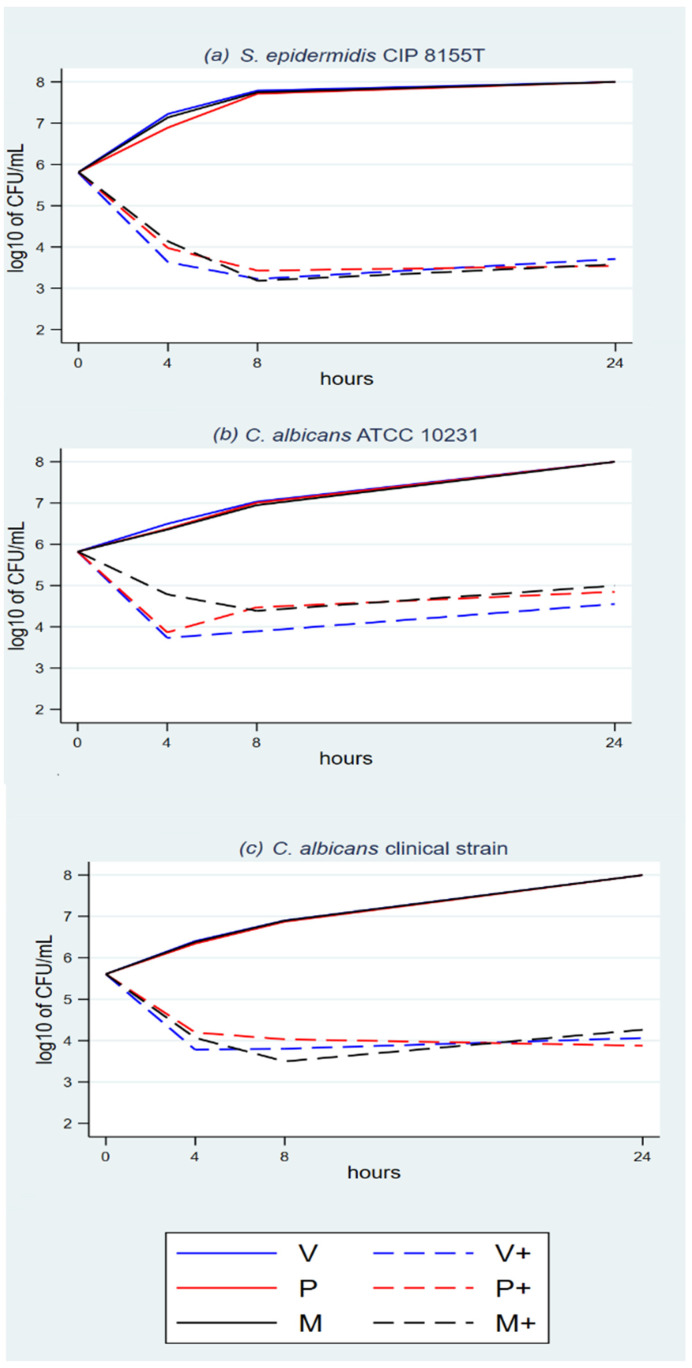
(**a**–**c**) Time-kill analyses *C. albicans* and *S. epidermidis* with the three TS.

**Figure 3 antibiotics-11-01195-f003:**
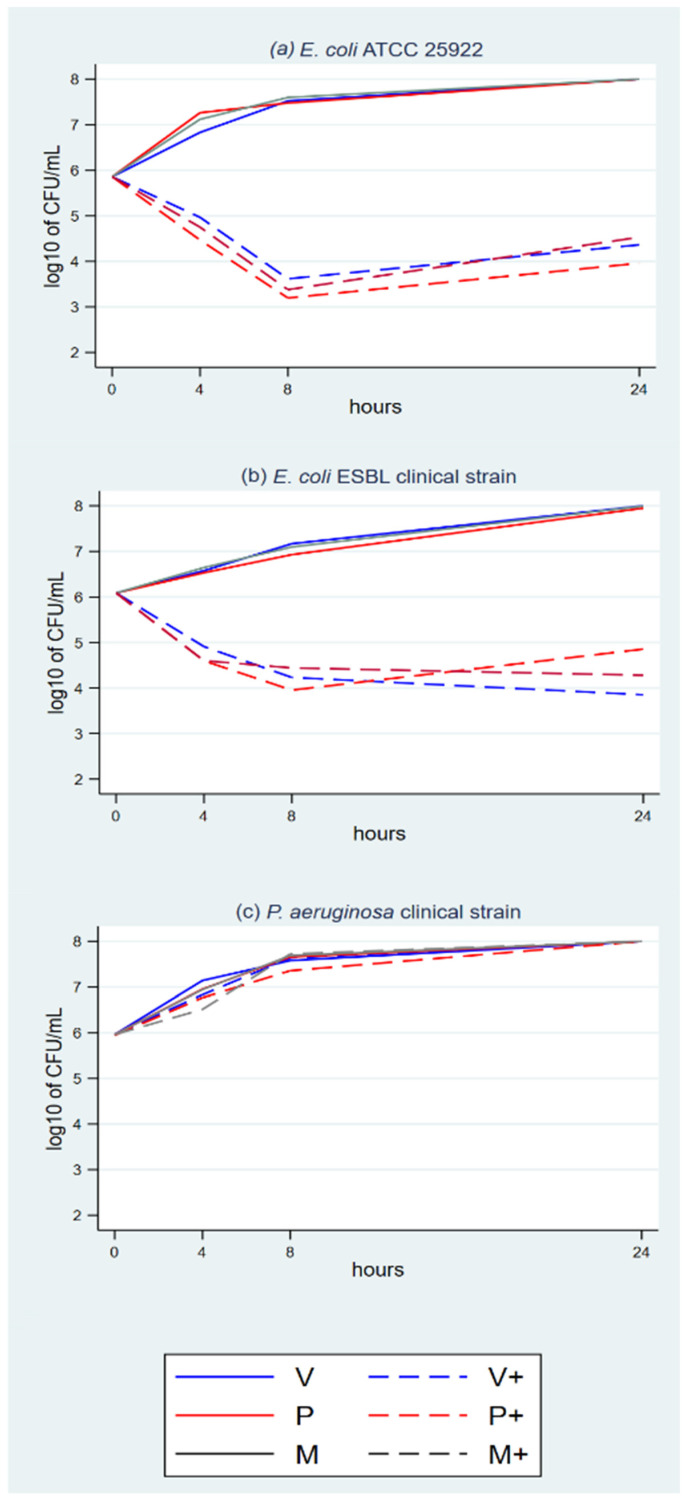
(**a**–**c**) Time-kill analyses *E. coli* and *P. aeruginosa* with the three TS.

**Figure 4 antibiotics-11-01195-f004:**
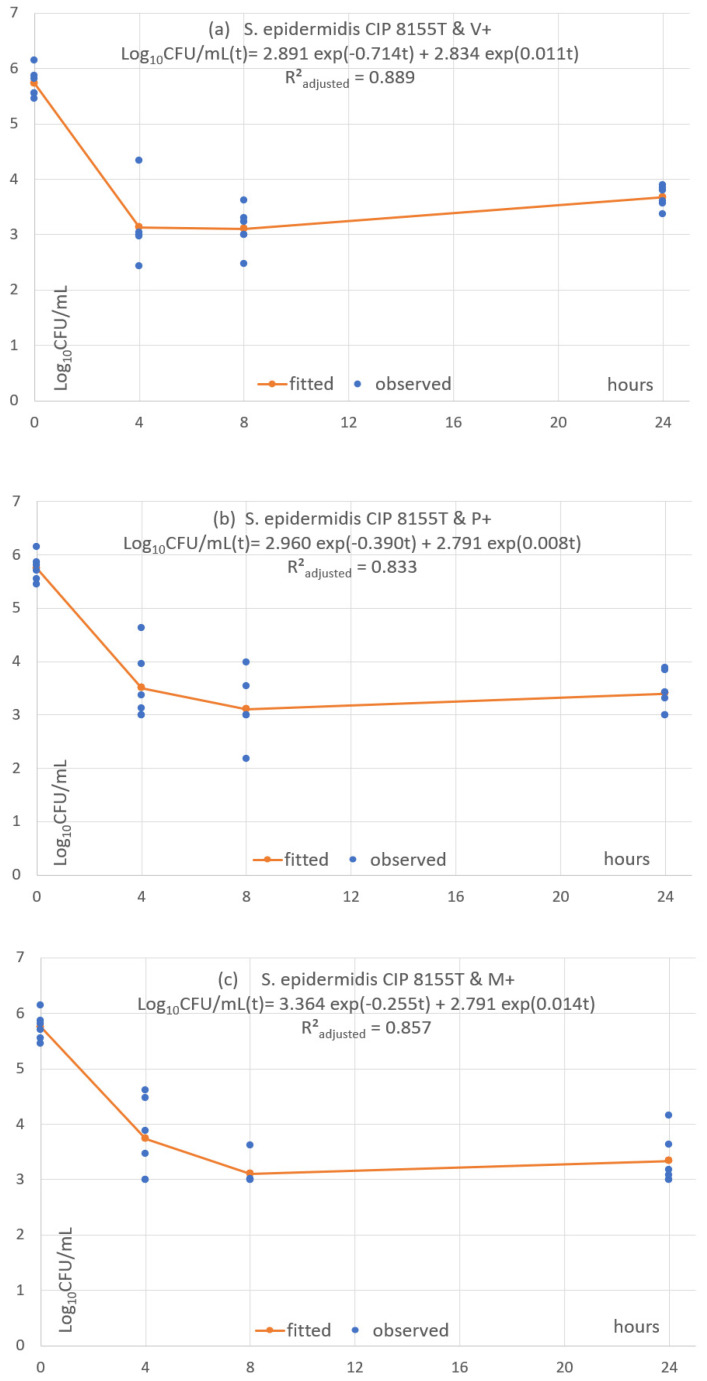
(**a**–**c**) Fitted predictive pharmacodynamic functions *S. epidermidis* with the three TS.

**Figure 5 antibiotics-11-01195-f005:**
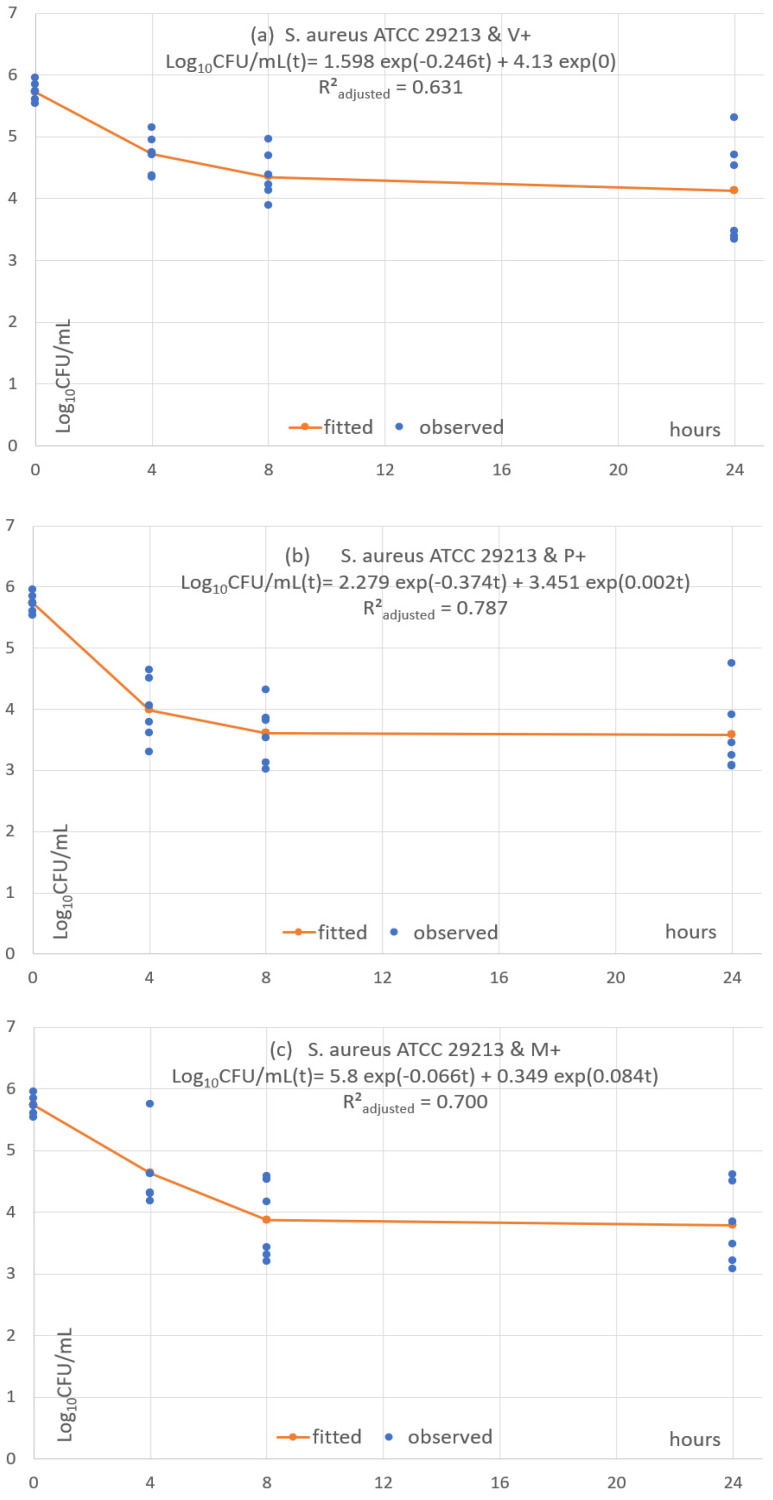
(**a**–**c**) Fitted predictive pharmacodynamic functions *S. aureus* with the three TS.

**Figure 6 antibiotics-11-01195-f006:**
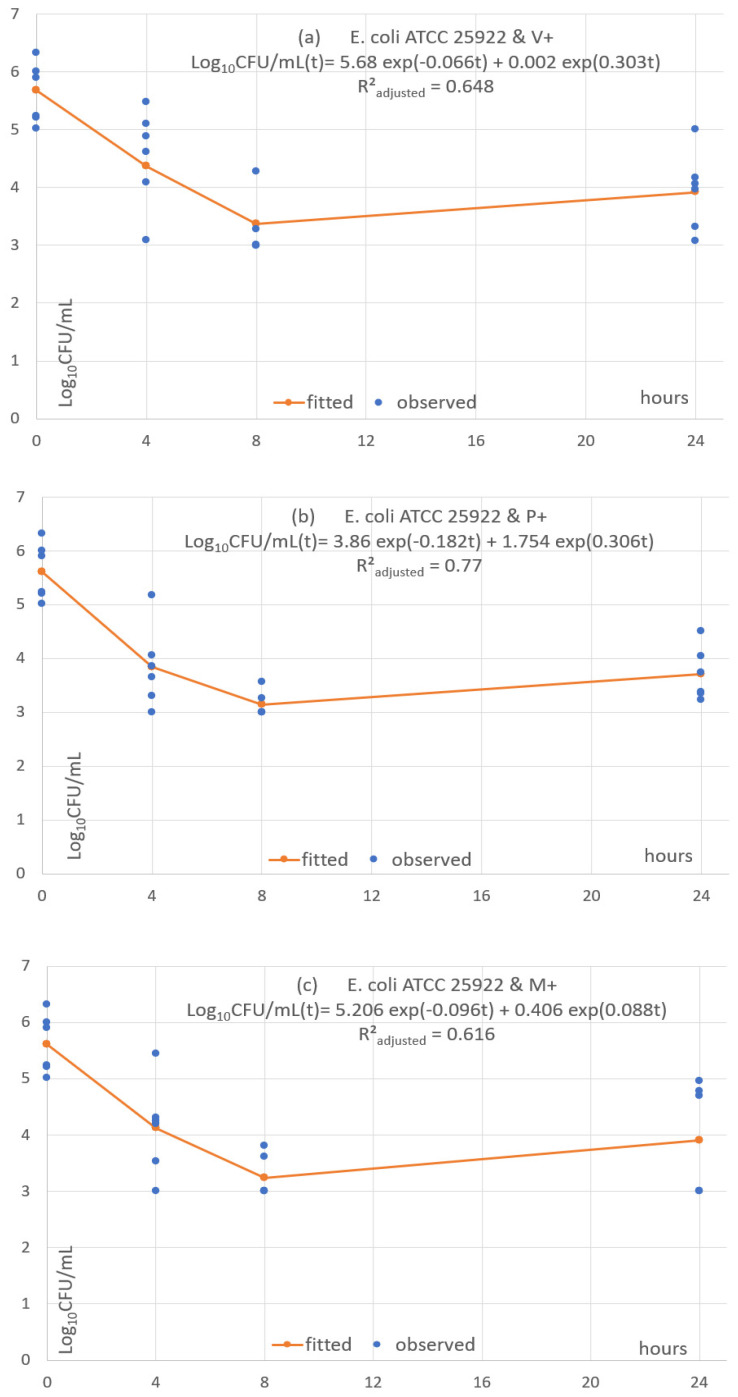
(**a**–**c**) Fitted predictive pharmacodynamic functions *E. Coli* with the three TS.

**Figure 7 antibiotics-11-01195-f007:**
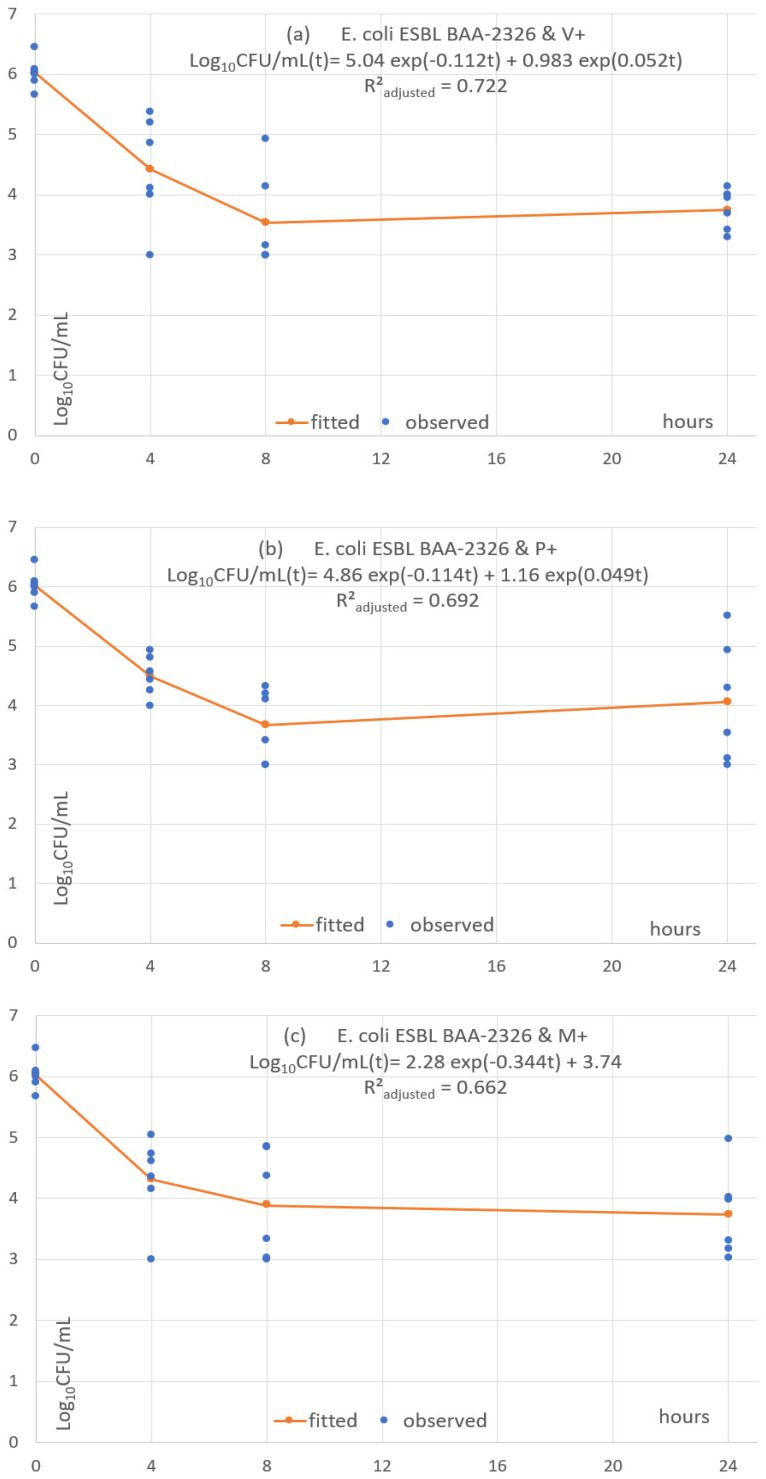
(**a**–**c**) Fitted predictive pharmacodynamic functions ESBL *E. Coli* with the three TS.

**Table 1 antibiotics-11-01195-t001:** Comprehensive OLS regression of log_10_ microbial concentration depending on triclosan, material, strain and timeslot.

log(CFU)	Coef.	St.Err.	*t*-Value	*p*-Value	95% Conf	Interval	Sig.
*S. epidermidis* CIP 8155T	0	Reference level for microorganisms					
*C. albicans* ATCC 10231	−0.097	0.124	−0.78	0.436	−0.339	0.147	
*C. albicans* clinical	−0.205	0.124	−1.66	0.098	−0.448	0.038	
*E. coli* ATCC 25929	0.052	0.124	0.42	0.674	−0.191	0.295	
*E. coli* ESBL clinical	0.183	0.124	1.48	0.140	−0.060	0.426	
MRSA ATCC 33592	0.152	0.124	1.23	0.221	−0.195	0.395	
MRSA clinical strain	0.048	0.124	0.38	0.700	−0.205	0.291	
*P. aeruginosa* clinical	1.306	0.124	10.55	0.000	1.063	1.549	*
*S. aureus* ATCC 29213	0.170	0.124	1.37	0.170	−0.073	0.413	
Polyglecaprone 25	0	Reference level for suture materials					
Polyglactin 910	0.035	0.072	0.47	0.637	−0.105	0.175	
Polydioxanone	−0.015	0.072	−0.21	0.835	−0.156	0.125	
No triclosan	0	Reference level for use of triclosan					
Triclosan	−2.256	0.058	−38.64	0.000	−2.370	−2.141	*
0 h	0	Reference level for hours					
4	−0.379	0.083	−4.66	0.000	−0.541	−0.217	*
8	−0.237	0.083	−2.90	0.005	−0.399	−0.075	**
24	0.416	0.083	5.11	0.000	0.254	0.578	*
Overall model intercept	4.400	0.113	38.92	0.000	4.179	4.622	*
Number of obs		1296		Number of groups		324	
F-test		df = 141,281		*p*-value		<0.0001	
R-squared		0.5847		Adjusted R-squared		0.5802	

* *p* < 0.001, ** *p* < 0.005, Sig.: significant, df: degrees of freedom, St. Err.: standard error. Hausman test: significance of: fails to meet the asymptotic assumptions. Breusch and Pagan Lagrangian multiplier test (LM) for random effects: *p*-value = 0.2085.

**Table 2 antibiotics-11-01195-t002:** Focused random effects panel regression of log_10_ microbial concentration depending on material, strain and timeslot.

log(CFU)	Coef.	St.Err.	*t*-Value	*p*-Value	95% Conf	Interval	Sig.
*S. epidermidis* CIP 8155T	0	Reference level for microorganisms					
*C. albicans* ATCC 10231	0.341	0.128	2.65	0.008	0.089	0.592	
*C. albicans* clinical	0.098	0.128	0.77	0.444	−0.153	0.350	
*E. coli* ATCC 25929	0.265	0.128	2.07	0.039	0.014	0.517	
*E. coli* ESBL clinical	0.551	0.128	4.30	0.000	0.210	0.803	*
MRSA ATCC 33592	0.537	0.128	4.19	0.000	0.286	0.789	*
MRSA clinical strain	0.221	0.128	1.72	0.085	−0.030	0.472	
*S. aureus* ATCC 29213	0.545	0.128	4.25	0.000	0.294	0.796	*
Polyglecaprone 25	0	Reference level for suture materials					
Polyglactin 910	0.071	0.079	0.90	0.369	−0.083	0.225	
Polydioxanone	−0.008	0.079	−0.11	0.915	−0.162	0.146	
0 h	0	Reference level for hours					
4	−1.795	0.063	−28.28	0.000	−1.919	−1.670	*
8	−2.148	0.063	−33.85	0.000	−2.273	−2.023	*
24	−1.936	0.063	−30.50	0.000	−2.060	−1.811	*
Overall model intercept	5.392	0.109	49.64	0.000	5.179	5.605	**
Number of obs		576		Number of groups		144	
Wald Chi-squared		df = 12		*p*-value		<0.0001	
R-squared		0.687					

*P. aeruginosa* and NTS not included in the focused model; * *p* < 0.001, ** *p* < 0.005, Sig.: significant, df: degrees of freedom, St. Err.: standard error. Hausman test: *p* = 1. Breusch and Pagan Lagrangian multiplier test (LM) for random effects: *p*-value < 0.0001.

**Table 3 antibiotics-11-01195-t003:** Fitting pharmacodynamic model (3) microbial concentrations from T0 to T24, excluding NTS and *P. aeruginosa*.

Microbial Strain	Suture	*R²_adjusted_*	C1 µg/mL	HF1 Hours	C2 µg/mL	HL2 Hours	Plot Shape
*S. aureus ATCC 29213*	V+	0.631	1.600	2.820	4.129	NA.	T0–T8: approx. −1.5log_10_ T8–T24: plateau & mild decrease
*S. aureus ATCC 29213*	P+	0.787	2.279	1.852	3.451	451	T0–T8: approx. −2log_10_ T8–T24: steady plateau
*S. aureus ATCC 29213*	M+	0.700	5.382	10.558	0.359	8.171	T0–T8: approx. −2log_10_ T8–T24: plateau & mild decrease
*MRSA ATCC 33592*	V+	0.676	2.542	2.633	3.445	98.943	T0–T8: approx. −2log_10_ T8–T24: plateau & mild increase
*MRSA ATCC 33592*	P+	0.710	2.696	1.974	3.691	210.724	T0–T8: approx. T8–T24: plateau & mild increase
*MRSA ATCC 33592*	M+	0.858	3.017	2.000	2.9700	64.903	T0–T8: approx. −2.5log_10_ T8–T24: plateau & mild increase
*MRSA clinical*	V+	0.700	2.070	0.633	3.776	NA.	T0–T4: approx. −2log_10_ T4–T24: steady plateau
*MRSA clinical*	P+	0.836	2.680	1.613	3.166	152.646	T0–T8: approx. −2.5log_10_ T8–T24: plateau & mild increase
*MRSA clinical*	M+	0.818	2.547	1.677	3.297	NA.	T0–T8: approx. −2.5log_10_ T8–T24: plateau & mild decrease
*S. epidermidis CIP 8155T*	V+	0.889	2.891	0.971	2.834	64.203	T0–T8: approx. −2.5log_10_ T8–T24: plateau & mild increase
*S. epidermidis CIP 8155T*	P+	0.833	2.960	1.778	2.790	83.821	T0–T8: approx. −2.5log_10_ T8–T24: plateau & mild increase
*S. epidermidis CIP 8155T*	M+	0.857	3.364	2.714	2.389	50.196	T0–T8: approx. −2.5log_10_ T8–T24: plateau & mild increase
*E. coli ATCC 25922*	V+	0.648	5.682	10.528	0.0019	2.287	T0–T8: approx. −2log_10_ T8–T24: plateau & mild increase
*E. coli ATCC 25922*	P+	0.771	3.858	3.791	1.754	22.728	T0–T8: approx. −2.5log_10_ T8–T24: plateau & mild increase
*E. coli ATCC 25922*	M+	0.616	5.206	7.196	0.406	7.884	T0–T8: approx. −2.5log_10_ T8–T24: plateau & mild increase
*E. coli ESBL BAA-2326*	V+	0.721	5.037	6.162	0.983	13.36	T0–T8: approx. −2.5log_10_ T8–T24: plateau & mild increase
*E. coli ESBL BAA-2326*	P+	0.692	4.857	6.091	1.162	14.216	T0–T8: approx. −2.5log_10_ T8–T24: plateau & mild increase
*E. coli ESBL BAA-2326*	M+	0.662	2.78	2.014	2.015	NA.	T0–T8: approx. −2log_10_ T8–T24: plateau & mild decrease
*C. albicans ATCC 10231*	V+	0.629	2.104	0.185	3.371	93.158	T0–T4: approx. −2log_10_ T4–T24: plateau & mild increase
*C. albicans ATCC 10231*	P+	0.494	1.864	0.166	3.611	86.005	T0–T4: approx. −1.7log_10_ T4–T24: plateau & mild increase
*C. albicans ATCC 10231*	M+	0.381	1.824	0.184	3.651	98.992	T0–T4: approx. −1.7log_10_ T4–T24: plateau & mild increase
*C. albicans clinical*	V+	0.647	5.673	10.400	0.006	2.723	T0–T8: approx. −2log_10_ T8–T24: plateau & mild increase
*C. albicans clinical*	P+	0.648	5.675	10.568	0.001	2.070	T0–T8: approx. −2log_10_ T8–T24: plateau & mild increase
*C. albicans clinical*	M+	0.648	5.680	10.534	0.001	2.157	T0–T8: approx. −2log_10_ T8–T24: plateau & mild increase

Note: “mild” means a variation of less than 1log_10_.

## Data Availability

All source data is in the Appendix A (raw data table).

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
