# Peer review of "Do Different Sutures with Triclosan Have Different Antimicrobial Activities? A Pharmacodynamic Approach"

_antibiotics, 2022, doi:10.3390/antibiotics11091195_

Round 1

Reviewer 1 Report

The authors evaluate the efficacy of Triclosan surgical sutures with different loads against 8 triclosan susceptible microorganisms. Such a study is valuable as it allows the determination of PD relationship and better selection of Triclosan load in absorbable sutures. The authors have performed in vitro time-kill assays by exposing the microorganisms to sutures. Although the experiments are interesting, the analysis is poorly performed and cannot be translated in anyway.

Major comments:

The time-kill curves are measured but the concentrations of Triclosan are not known and bridging these two to derive PK/PD relation is key. Hence, authors should measure the Triclosan levels in the culture and if possible, the amount (or dose) remaining the suture. Like what they have done in their previous study published in Pharmaceutics https://doi.org/10.3390/pharmaceutics14030539

Page 6 Line 198: How can microbial concentration be associated with the presence of TS without measuring TS levels? The statistical analysis does not make sense. Why was this statistical analysis done? What do the authors want to show? Why was OLS regression performed? Kindly explain.

The data representation is being repeated for no reason. For example, Figure 1 and Figure 4.

The 3 sutures showed no difference – because they release the same amount of TS or the amount is well above MIC?

What are the PK indices that drives antimicrobial activity of TS? Is it same for all the 8 microorganisms?

The authors conclude a significant reduction of microbial dose close to the sutures f implantation – here is no data in the manuscript that supports that conclusion. How was this analysis performed and conclusion derived? Kindly explain.

The current study does not add any value due to poor analysis and cannot be translated to clinic. The authors conclude no significant difference in 3 sutures at different levels of TS load.

A simple logical question to ask, is why these sutures at different loads even manufactured and used in the clinic if all have the same antimicrobial activity. Clearly, the in vitro study and analysis performed is weak/limited and does not provide any insights to derive any conclusion.

Minor Comments:

Page 3, Line 111: Why is 35 cm an optimal TS length to assess antimicrobial activity? Kindly explain or provide a reference

The time-kill figures are of poor quality.

Mention the TS concentrations in V+, P+ and M+ on the figure legend similar to “V+, 472 ug/m TS” or in the figure caption.

There is no positive control for antimicrobial kill.

Reviewer 2 Report

Congratulations for this interesting paper. The use of triclosan in the sutures surely is a matter of interest for many surgeon.

Some surgical activity are today discussed for the intensive use of suture: in a paper like this the use of sutureless techinique and postoperative infection should be discussed in the introduction section.

Please add a short paragraph in the introduction section related to the use of suture in surgery and cite the following:

Chisci G, Capuano A, Parrini S. Alveolar Osteitis and Third Molar Pathologies. J Oral Maxillofac Surg. 2018 Feb;76(2):235-236.

Chisci G, Parrini S, Capuano A. The use of suture-less technique following third molar surgery. Int J Oral Maxillofac Surg. 2013 Jan;42(1):150-1.

Author Response

Paragraph added as requested:

There is a broad array surgical wound closure methods, including thousands of suture types, staples and surgical adhesives [1]. Sutureless surgery is also being developed in various fields, including maxillofacial and cardiac surgery [2-5]. Minimising the risk of surgical site infection (SSI) is an important consideration in the development of surgical wound closure techniques.

Reviewer 3 Report

It’s a nice article for the purpose it is written for which is the inhibition of gram-positive and gram-negative bacteria (except pseudomonas) at surgical sites.  

The title and abstract are appropriate for the content of the text. The article is well constructed and clearly written. The methods are clearly described, and the results are supported by enough references and statistical analysis.

Notes regarding the article:

1- Lines 50-51 : However, triclosan’s low aqueous solubility limits its bioavailability

comment: If the author means by "bioavailability” after dermal application or skin exposure to triclosan, then the low bioavailability might be advantageous as triclosan is considered to be safe when used in topical preparations, or consumer items intended to reduce or prevent bacterial contamination, skin exposure to triclosan.

At this point, the author (18) referenced to an article that discussed the enhancement of the availability and substantivity of triclosan through cyclodextrin complexation.

Please clarify

2.   Since the diffusion in an aqueous solution is low, why triclosan was not solubilized in different concentrations in a solvent such as DMSO or others to determine the MIC of this substance in vitro?

3.   In the time-kill curve, there was growth after 8 hours of exposure, if TS becomes inactive with time, should be indicated in the introduction.

Author Response

Comments and Suggestions for Authors

It’s a nice article for the purpose it is written for which is the inhibition of gram-positive and gram-negative bacteria (except pseudomonas) at surgical sites.  

The title and abstract are appropriate for the content of the text. The article is well constructed and clearly written. The methods are clearly described, and the results are supported by enough references and statistical analysis.

Notes regarding the article:

1- Lines 50-51 : However, triclosan’s low aqueous solubility limits its bioavailability

comment: If the author means by "bioavailability” after dermal application or skin exposure to triclosan, then the low bioavailability might be advantageous as triclosan is considered to be safe when used in topical preparations, or consumer items intended to reduce or prevent bacterial contamination, skin exposure to triclosan.

At this point, the author (18) referenced to an article that discussed the enhancement of the availability and substantivity of triclosan through cyclodextrin complexation.

Please clarify

Paragraph deleted: It was intended to provide background information on the methods required for triclosan to reach target microoganisms in a water solution. It does not support the justification of this study where triclosan is used in its solid state.

  1.  Since the diffusion in an aqueous solution is low, why triclosan was not solubilized in different concentrations in a solvent such as DMSO or others to determine the MIC of this substance in vitro?

The microbiology lab determined its method to verify the in vitro triclosan MIC of the studied species and I checked that their findings were within published ranges.

  1. In the time-kill curve, there was growth after 8 hours of exposure, if TS becomes inactive with time, should be indicated in the introduction.

 Added paragraph: The two main triclosan degradation pathways are hydrolysis and photodegradation [8, 9]. Degradation in water is too slow to be significant over 24 hours, providing it is not exposed to intense light.

Round 2

Reviewer 1 Report

I thank the authors for providing the response to my comments. I also thank for providing higher quality images. Unfortunately, my major concerns are still not addressed, and the manuscript falls short at several levels. Deriving clinical conclusions based on inconclusive experimentation could have fatal outcomes.

The Triclosan release kinetics have not been measured in this study, the authors could have used the release kinetics from their previous study to estimate the concentration levels to further link with the PD.

The authors also mentioned in their response that Triclosan (release) concentrations are not required to identify that Triclosan concentrations are in sufficient amounts to kill/eradicate susceptible microorganisms. Considering that all the 3 sutures show no difference probably because, it is even more important to estimate concentrations. It could be possible that the Triclosan concentrations are very high.

Further, the authors mention (in their response) that in vitro assay and quantitative results cannot be translated to humans or large animals – thus making this study redundant. In such a case, the authors need to design a study in such a way that clinical transformation is possible or atleast provide some fundamental information to design/feed into a translational understanding. One could even critique by saying “are these sutures for clinical use or in vitro use?”

It is inaccurate to say that all 3 sutures have no significant differences in activity. Such a conclusion into clinic will be misleading and dangerous. The manuscript essentially suggests that all clinicians that irrespective of the level of infection, if the microorganism is susceptible to Triclosan, a suture containing low dose is more than enough.

It is interesting that authors explain the MIC for various microorganisms but do not link it to PD in the study. The authors also mention that all 3 sutures have very different mechanical and bio-absorption properties. The study fails as the authors did not present the Triclosan release profiles and depend on a non-translatable in vitro assay to infer clinical results.

The authors have not included a positive control.

The authors infer suggestions for RCT’s i.e., two or more types of sutures should offer similar antimicrobial effect. Why is there a need to use two or more sutures if they have similar activities?

Round 3

Reviewer 1 Report

I thank the author addressing my comments and providing some background information.

My suggestion to move forward and fully applicable (especially after considering the scope of the study) is to explicitly mention that these findings/conclusions are derived based on in vitro testing. For example, kindly consider revising the below,

Title to “Do different sutures with triclosan have different in vitro antimicrobial activities? A pharmacodynamic approach.”

Line 14: “The purpose of this in vitro study is to ” or "in vitro pharmacodynamics"

Line 25: “No significant in vitro antimicrobial pharmacodynamic difference between the three TS is identified.”

Line 91: “analyze in vitro pharmacodynamics”

Line 257: “first in vitro pharmacodynamics study”

Line 317: re-write sentence by mentioning in vitro

Line 331: Which antimicrobial effect is over estimated – in vitro or in vivo.

Line 353: “This in vitro study/assessment shows that triclosan”

Line 354: Revise sentence. For example, “The antimicrobial profiles of (names of the suture) sutures present no in vitro difference….”

Line 356: “In vitro pharmacodynamics suggest….” Or “Based/Under the in vitro assessed/tested/experimental conditions, pharmacodynamics suggest….“

Author Response

Dear Reviewer,

We agreed with all changes and implemented them.

Best regards

The authors

Line 14: “The purpose of this in vitro study is to ” or "in vitro pharmacodynamics" (done)

Line 25: “No significant in vitro antimicrobial pharmacodynamic difference between the three TS is identified.” (done)

Line 91: “analyze in vitro pharmacodynamics” (done)

Line 257: “first in vitro pharmacodynamics study” (done)

Line 317: re-write sentence by mentioning in vitro (done)

Line 331: Which antimicrobial effect is over estimated – in vitro or in vivo. (done)

Line 353: “This in vitro study/assessment shows that triclosan” (done)

Line 354: Revise sentence. For example, “The antimicrobial profiles of (names of the suture) sutures present no in vitro difference….” (done)

Line 356: “In vitro pharmacodynamics suggest….” Or “Based/Under the in vitro assessed/tested/experimental conditions, pharmacodynamics suggest….“ (done)